# Association of Serum Fatty Acids at Admission with the Age of Onset of Acute Ischemic Stroke

**DOI:** 10.3390/nu12082411

**Published:** 2020-08-12

**Authors:** Takahisa Mori, Kazuhiro Yoshioka, Yuhei Tanno, Shigen Kasakura

**Affiliations:** Department of Stroke Treatment, Shonan Kamakura General Hospital, Okamoto 1370-1, Kamakura City, Japan; y.kazuhiro12@icloud.com (K.Y.); yxip01@icloud.com (Y.T.); kasakura@med.kitasato-u.ac.jp (S.K.)

**Keywords:** serum fatty acids, admission, acute ischemic stroke, age of onset, dihomo-gamma-linolenic acid, docosahexaenoic acid

## Abstract

Dietary triglycerides influence fatty acid (FA) serum concentrations and weight percentages (wt %), which may be associated with the age of onset of acute ischemic stroke (AIS). We investigated the correlations between serum FA levels and proportions at admission and the age of onset of AIS. We evaluated patients with AIS admitted between 2016 and 2019 within 24 h of AIS onset and calculated the correlation coefficients between their ages, serum FA concentrations, and FA wt % values. Multiple linear regression analysis was performed to identify independent FAs indicating AIS age of onset. Furthermore, we estimated the threshold values of independent FAs for age of onset <60 years using receiver operating characteristic curves by logistic regression. A total of 525 patients (median age: 75 years) met the inclusion criteria. The concentration of dihomo-gamma-linolenic acid (DGLA) and wt % of docosahexaenoic acid (DHA) were significant independent variables for age of onset of AIS, and receiver operating characteristic curves for age of onset <60 years showed thresholds of ≥117.7 µmol/L for DGLA and ≤3.7% for DHA. An increased DGLA concentration and decreased DHA wt % were significantly associated with onset of AIS at a younger age.

## 1. Introduction

Elevated serum triglycerides (TGs), which consist of glycerol and three fatty acids (FAs), are a risk factor for vascular disease [1]. The incidence of ischemic vascular disease increases with increasing age. Dietary TGs, which are derived from meat, fish, and vegetables, influence serum TG and FA levels [2]. Higher consumption of fish and n-3 polyunsaturated fatty acids (n-3 PUFAs) is associated with a reduced risk of thrombotic infarction [3] and coronary artery disease (CAD) [4] in healthy middle-aged subjects. Previous studies have reported the neuroprotective effects of docosahexaenoic acid (DHA) [5,6,7]. The relationship between serum FA levels under fasting conditions and stroke has been reported [8,9,10]. However, few studies have reported on serum FA levels at the onset of ischemic events [11,12], which are typically examined under non-fasting conditions and associated with the dietary intake of TGs. Higher intake of seafood may be associated with higher n-3 PUFA levels, which is likely to reduce the risk of ischemic events.

A low serum n-3 PUFA/n-6 PUFA ratio may be associated with neurological deterioration in patients with acute ischemic stroke (AIS) [13], and low serum n-3 PUFA levels may also be associated with the onset of AIS.

The weight percentages (wt %) of individual FAs, with respect to total serum FAs, have been examined in healthy humans. With increasing age, the wt % of n-6 PUFAs and n-3 PUFAs decrease and increase, respectively [14]. CAD results in increased serum concentrations of palmitic acid (PA), stearic acid (StA), oleic acid (OlA), linoleic acid (LiA), and arachidonic acid (AA) and a lower serum concentration of eicosapentaenoic acid (EPA) compared to subjects without CAD. Additionally, the wt % values of EPA and DHA are lower in subjects with CAD than in healthy controls [15]. Interestingly, in patients who experienced lacunar or atherosclerotic stroke between the ages of 50–74 years, the serum concentrations of saturated fatty acids (SFAs), n-9 monounsaturated fatty acid (n-9 MUFA), and n-6 PUFAs are elevated [11,12]. This suggests that the serum concentrations and wt % of FAs are related to the age of acute ischemic disease onset. Dietary DHA has beneficial effects on neurodevelopment in mice [16]. DHA and AA are rich in phospholipids and are found in the central nervous system [17]. In this study, we investigated the associations between the serum concentrations and proportions of FAs in patients admitted for AIS and age of AIS onset.

## 2. Materials and Methods

We conducted a cross-sectional study of patients with AIS who were admitted to our institution between August 2016 and July 2019 within 24 h of AIS onset and underwent evaluation of blood lipids and FAs at admission. We excluded patients with a pre-hospital modified Rankin scale score ≥3 or body mass index <18.5, who were defined as having a severe disability, or who were underweight, respectively, according to World Health Organization guidelines. This was performed to exclude patients with possible malnutrition.

### 2.1. Measurement of Serum Lipids and FAs

Serum total cholesterol (T-CHO), TGs, and high-density lipoprotein-cholesterol (HDL-C) were measured enzymatically using reagents manufactured by Denka Seiken (Denka Co., Ltd., Chuo, Tokyo, Japan) on a BioMajesty 6050 High Throughput Clinical Chemistry Analyzer (JEOL Ltd., Akishima, Tokyo, Japan). Low-density lipoprotein-cholesterol (LDL-C) was calculated using the Friedewald formula as follows: LDL-C = (T-CHO − HDL-C − TG)/5. We examined the SFAs lauric acid (LaA), myristic acid (MyA), PA, and StA; the n-9 MUFA OlA; the n-6 PUFAs LiA, dihomo-gamma-linolenic acid (DGLA), and AA; and the n-3 PUFAs alpha-linolenic acid (AlA), EPA, and DHA. We measured the serum concentrations and wt % of each FA at admission. The FAs in 1 mL of serum were measured. FAs were extracted according to a general technique using tricosanoic acid (Nu-Chek Prep, Inc., Elysian, MN, USA) as an internal standard. Lipid extracts were hydrolyzed, extracted with chloroform, and dried under nitrogen gas. After adding 30% potassium methoxide methanol solution to the residual sample, the sample was incubated at 100 °C for 5 min and cooled. Samples were extracted with hexane and analyzed on a GC-2010 Plus Capillary Gas Chromatograph (Shimadzu, Kyoto, Japan) equipped with a flame ionization detector, using a BPX70 column. The operating conditions were as follows: 50 °C for 0.5 min, temperature increase to 260 °C over 25 min, and holding at this temperature for 5 min. The injector and detector temperatures were 240 °C and 280 °C, respectively, and helium was used as the carrier gas at 1.09 mL/min. Component identification was performed by comparing the retention times of the samples with those of standards. We evaluated the concentrations of serum lipids and FAs and wt % of serum FAs using internal standard ratios.

### 2.2. Ethical Approval and Consent to Participate

All procedures were performed in accordance with the ethical standards of the institution and the 1964 Helsinki Declaration. The Tokusyukai Group Ethical Committee approved our retrospective study (TGE01486-024).

### 2.3. Consent to Participate

Written informed consent for participation and publication was not required. The study was based on an opt-out model of enrolment, which was permitted by the ethical committee.

### 2.4. Statistical Analysis

We expressed non-normally distributed continuous variables as medians and interquartile ranges. We used a multiple comparison test to compare all possible FA pairs and Spearman rank correlation coefficient (*r*_s_) to measure the strength of the relationships between non-normally distributed variables. We defined 0 ≤ |*r*_s_| < 0.1 as no correlation, 0.1 ≤ |*r*_s_| < 0.4 as a weak correlation, 0.4 ≤ |*r*_s_| < 0.6 as a moderate correlation, and 0.6 ≤ |*r*_s_| as a strong correlation. Multicollinearity was defined as a strong correlation between variables. After excluding FAs with multicollinearity, we performed multiple linear regression analysis to identify independent FAs affecting the AIS age of onset. Multiple logistic regression analysis was also used to identify independent FAs correlated with ages of onset of <60 and <80 years. We estimated the threshold values of FAs for onset at age <60 or <80 years using area under the curve (AUC) values derived from the receiver operating characteristic (ROC) curves of the logistic regression model. A probability *p* < 0.05 was considered as statistically significant. We used the JMP software (version 15.1; SAS Institute, Cary, NC, USA) for all statistical analyses.

## 3. Results

A total of 880 patients with AIS were admitted to our stroke center during the study period, and 525 patients met our inclusion criteria. Patient characteristics are summarized in Table 1 AIS occurred in 61 patients (11.6%) aged <60 years and in 341 patients (65.0%) aged <80 years. Age was negatively correlated with most FA concentrations and FA wt % values, except for n-3 PUFAs (Table 2).

After excluding FA concentrations with multicollinearity and those without a correlation with age, we performed multiple linear regression analysis using the MyA, LiA, DGLA, AA, and EPA concentrations as explanatory variables. The results revealed that LiA and DGLA were independent variables when the age at AIS onset was an objective variable (Table 3). We also performed multiple linear regression analysis using the LaA, OlA, DGLA, and DHA wt % values as explanatory variables and found that DGLA wt % and DHA wt % were independent variables when the age ofAIS onset was an objective variable (Table 3).

Next, after excluding the DGLA wt % because of multicollinearity (Table 2), we performed multiple linear regression analysis using the LiA and DGLA concentrations and DHA wt %, which demonstrated that the DGLA concentration and DHA wt % were independent variables when age at AIS onset was an objective variable (Table 3). Multiple logistic regression analysis showed that DGLA concentration and DHA wt % were significant variables when the objective variable was age of onset <60 years (AUC: 0.810) and <80 years (AUC: 0.648). The ROC curves estimated threshold values for the DGLA concentration and DHA wt % for onset at age <60 and <80 years are shown in Table 4.

## 4. Discussion

Our results demonstrate that in patients with AIS, the DGLA concentration and DHA wt % are significant independent variables for the age of AIS onset and that increased DGLA concentrations and decreased DHA proportions are significantly associated with young-onset AIS.

Except for DHA, FA concentrations were positively correlated with the age of AIS onset; the levels of LiA and DGLA (n-6 PUFAs) were strongly correlated with the levels of PA and StA (SFAs), as well as OlA (an n-9 MUFA). There were strong correlations between various SFA concentrations. The OlA concentration was strongly correlated with the levels of SFAs, LiA, and DGLA (n-6 PUFAs), and AlA (n-3 PUFA). The EPA concentration was strongly correlated with that of DHA (Table 2). These strong mutual correlations were related to the ingestion of certain foods, including large amounts of various FAs, as recent dietary quantities influenced serum concentrations at admission. The wt % values of FAs, which reflected the dietary FA composition, may be influenced not by the quantity of ingested FAs but rather by their quality. The wt % values of five FAs (LaA, OlA, DGLA, EPA, and DHA) were correlated with the age of AIS onset, and EPA wt % was strongly correlated with the DHA wt %. FA wt % values displayed quite different trends from FA concentrations.

A previous study reported that elevated serum PA and OlA levels are associated with an increased frequency of incidental lacunar stroke and that elevated serum DHA and AA levels are associated with a decreased incidence of ischemic stroke [9]. However, the results were obtained in a case-controlled study, in which blood was collected under fasting conditions from all participants, and serum FA levels were not assessed at admission at AIS onset. In our series, the concentrations of PA, OlA, and AA at admission showed weak negative correlations with the age of AIS onset, although they were not independent variables.

In a study of healthy Canadians in their 20 s, the median concentrations of LiA, DGLA, and DHA were 2208.2, 68.2, and 82.0 μmol/L, respectively [18], which are lower than those detected in our study. Particularly, the median DHA concentration in our study was 376.7 μmol/L, which is much higher than that in healthy young people. In a study of patients with CAD in the United States with an average age of 47 years, the wt % of LiA, AA, EPA, and DHA were 58.6%, 9.4%, 0.9%, and 0.5%, respectively [15]. The proportions of LiA and AA were higher than those in the US patients compared to in our patients; however, the proportions of EPA, and DHA were lower. The median age in our study was 75 years, and 35% of patients were aged 80 years or older. The DHA wt % in our patients was much higher than that in the US patients (4.6% vs. 0.5%).

It has been suggested that n-3 PUFAs can reduce the incidence of CAD and stroke, as well as mortality associated with cardiovascular disease [19,20,21,22,23]. Administration of purified EPA reduced recurrent stroke [21]. In contrast, SFAs may increase the risk of these conditions [24]. In our patients, the concentrations and proportions of SFAs and EPA were not independent variables for the age of AIS onset; however, DHA wt % was an independent variable, and elevated DHA wt % values were associated with late-onset AIS.

Dietary intake of TGs, which are found in meat, fish, vegetables, and their oils, influences serum FA concentrations and proportions. Therefore, it is essential to assess the types and quantities of dietary TG sources consumed during the days prior to AIS onset. Dietary intake assessment and developing dietary regimens that decrease the DGLA concentration and increase the DHA proportion may prevent AIS or delay its onset. A prospective randomized control study is required to identify the appropriate intake of FAs to achieve these goals.

## Figures and Tables

**Table 1 nutrients-12-02411-t001:** Patient characteristics.

	*n* = 525
Age (MD, IQR) (min, max) years	75 (68–81) (min: 39, max: 99)
Male sex	337 (64.2%)
Height (MD, IQR) cm	163 (155–168)
Body weight (MD, IQR) kg	60 (53–68.5)
Body mass index (MD, IQR) kg/m^2^	23.1 (21.0–25.4)
**Glucose and lipids**	
Blood sugar (MD, IQR) mmol/L	6.83 (5.94–8.49)
Hemoglobin A1c (MD, IQR)% (NGSP)	5.9 (5.6–6.4)
Total cholesterol (MD, IQR) mmol/L	5.09 (4.42–5.84)
High-density lipoprotein cholesterol (MD, IQR) mmol/L	1.43 (1.19–1.74)
Triglycerides (MD, IQR) mmol/L	1.19 (0.81–1.80)
**Saturated fatty acids**	
Lauric acid (LaA) (MD, IQR) μmol/L	4.49 (2.99–8.48)
Myristic acid (MyA) (MD, IQR) μmol/L	70.08 (52.12–97.67)
Palmitic acid (PA) (MD, IQR) μmol/L	2,472 (2,109–2,903)
Stearic acid (StA) (MD, IQR) μmol/L	638.5 (544.9–747.3)
LaA (MD, IQR)%	0 (0–0.1)
MyA (MD, IQR)%	0.6 (0.5–0.8)
PA (MD, IQR)%	23.6 (22.6–24.5)
StA (MD, IQR)%	6.7 (6.2–7.2)
**n-9 MUFA**	
Oleic acid (OlA) (MD, IQR) μmol/L	2,008 (1,656–2,527)
OlA (MD, IQR)%	21.1 (19.4–23.2)
**n-6 PUFAs**	
Linoleic acid (LiA) (MD, IQR) μmol/L	2,515 (2,150–2,942)
Dihomo-gamma-linolenic acid (DGLA) (MD, IQR) μmol/L	89.98 (69.76–115.57)
Arachidonic acid (AA) (MD, IQR) μmol/L	517.9 (439.0–604.5)
LiA (MD, IQR)%	26.1 (23.4–28.4)
DGLA (MD, IQR)%	1 (0.8–1.2)
AA (MD, IQR)%	5.9 (4.9–6.7)
**n-3 PUFAs**	
Alpha-linolenic acid (AlA) (MD, IQR) μmol/L	65.34 (49.00–87.42)
Eicosapentaenoic acid (EPA) (MD, IQR) μmol/L	206.2 (142.2–302.70)
Docosahexaenoic acid (DHA) (MD, IQR) μmol/L	376.66 (303.54–469.38)
AlA (MD, IQR)%	0.7 (0.6–0.8)
EPA (MD, IQR)%	2.2 (1.6–3.4)
DHA (MD, IQR)%	4.6 (3.7–5.5)
EPA/AA ratio (MD, IQR)	0.39 (0.265–0.575)
n-6/n-3 ratio (MD, IQR)	4.16 (3.2–5.315)

Min: minimum, max: maximum, MD: median, IQR: interquartile range; NGSP: National Glycohemoglobin Standardization Program; wt %: weight percentage of total fatty acids; n-3 PUFA: n-3 polyunsaturated fatty acid; n-6 PUFA: n-6 polyunsaturated fatty acid; n-9 MUFA: n-9 monounsaturated fatty acid.

**Table 2 nutrients-12-02411-t002:** Spearman correlation coefficients.

**Correlation Coefficients Between Age of AIS Onset and FA Concentrations**												
***r*_s_**	**Age**	**LaA**	**MyA**	**PA**	**StA**	**OlA**	**LiA**	**DGLA**	**AA**	**AlA**	**EPA**	**DHA**
**Age**		−0.18	−0.19	−0.22	−0.28	−0.25	−0.25	−0.33	−0.21	−0.16	0.11	0.05
**LaA**	−0.18		**0.81**	0.50	0.56	0.50	0.43	0.42	0.05	0.52	−0.06	0.01
**MyA**	−0.19	**0.81**		**0.74**	**0.74**	**0.72**	0.55	0.57	0.18	**0.69**	0.06	0.25
**PA**	−0.22	0.50	**0.74**		**0.80**	**0.91**	**0.72**	**0.62**	0.48	**0.66**	0.11	0.45
**StA**	−0.28	0.56	**0.74**	**0.80**		**0.73**	**0.70**	**0.63**	0.46	**0.66**	0.16	0.40
**OlA**	−0.25	0.50	**0.72**	**0.91**	**0.73**		**0.66**	**0.63**	0.40	**0.65**	−0.04	0.30
**LiA**	−0.25	0.43	0.55	**0.72**	**0.70**	**0.66**		0.49	0.34	**0.71**	−0.11	0.19
**DGLA**	−0.33	0.42	0.57	**0.62**	**0.63**	**0.63**	0.49		0.46	0.42	−0.23	0.06
**AA**	−0.21	0.05	0.18	0.48	0.46	0.40	0.34	0.46		0.18	0.10	0.32
**AlA**	−0.16	0.52	**0.69**	**0.66**	**0.66**	**0.65**	**0.71**	0.42	0.18		0.10	0.30
**EPA**	0.11	−0.06	0.06	0.11	0.16	−0.04	−0.11	−0.23	0.10	0.10		**0.70**
**DHA**	0.05	0.01	0.25	0.45	0.40	0.30	0.19	0.06	0.32	0.30	**0.70**	
**Correlation Coefficients Between Age of AIS Onset and FA Percentages**												
***r*_s_**	**Age**	**LaA%**	**MyA%**	**PA%**	**StA%**	**OlA%**	**LiA%**	**DGLA%**	**AA%**	**AlA%**	**EPA%**	**DHA%**
**Age**		−0.11	−0.10	−0.01	−0.07	−0.16	0.00	−0.23	0.00	−0.02	0.21	0.27
**LaA%**	−0.11		**0.68**	0.23	0.17	0.20	−0.07	0.10	0.34	0.32	0.14	−0.28
**MyA%**	−0.10	**0.68**		0.23	0.17	0.31	−0.20	0.22	−0.45	0.42	−0.15	−0.23
**PA%**	−0.01	0.23	0.23		−0.36	0.38	−0.46	0.08	−0.20	−0.13	−0.16	−0.08
**StA%**	−0.07	0.17	0.17	−0.36		−0.35	0.08	0.16	0.07	0.03	0.13	0.04
**OlA%**	−0.16	0.20	0.31	0.38	−0.35		−0.41	0.13	−0.34	0.10	−0.47	−0.41
**LiA%**	0.00	−0.07	−0.20	−0.46	0.08	−0.41		−0.04	−0.10	0.19	−0.30	−0.29
**DGLA%**	−0.23	0.10	0.22	0.08	0.16	0.13	−0.04		0.14	−0.07	−0.41	−0.36
**AA%**	0.00	0.34	−0.45	−0.20	0.07	−0.34	−0.10	0.14		−0.46	0.14	0.18
**AlA%**	−0.02	0.32	0.42	−0.13	0.03	0.10	0.19	−0.07	−0.46		−0.07	−0.12
**EPA%**	0.21	0.14	−0.15	−0.16	0.13	−0.47	−0.30	−0.41	0.14	−0.07		**0.75**
**DHA%**	0.27	−0.28	−0.23	−0.08	0.04	−0.41	−0.29	−0.36	0.18	−0.12	**0.75**	
**Correlation Coefficients Between the Four Significant Variables**												
***r*_s_**	**LiA**	**DGLA**	**DGLA%**	**DHA%**								
**LiA**		0.50	0.06	−0.39								
**DGLA**	0.50		**0.79**	−0.41								
**DGLA%**	0.06	**0.79**		−0.36								
**DHA%**	−0.39	−0.41	−0.36									

AA: arachidonic acid; AIS: acute ischemic stroke; AlA: alpha-linolenic acid; DHA: docosahexaenoic acid; DGLA: dihomo-gamma-linolenic acid; EPA: eicosapentaenoic acid; LaA: lauric acid; LiA: linoleic acid; MyA: myristic acid; OlA: oleic acid; PA: palmitic acid; StA: stearic acid.

**Table 3 nutrients-12-02411-t003:** Multiple linear regression analysis of the age of AIS onset.

**Between FA concentrations**			
Variables	*t*-value	*p*	adjusted *R*^2^
Myristic acid	1.43	ns	0.142
Linoleic acid (LiA)	−2.63	**<0.01**	
Dihomo-gamma-linolenic acid (DGLA)	−4.68	**<0.0001**	
Arachidonic acid	−1.63	ns	
Eicosapentaenoic acid	0.55	ns	
**Between FA percentages**			
Variables	*t*-value	*p*	adjusted *R*^2^
Lauric acid wt %	0.87	ns	0.119
Oleic acid wt %	−1.42	ns	
DGLA wt %	−3.66	**<0.001**	
Docosahexaenoic acid (DHA) wt %	4.75	**<0.0001**	
**Between LiA, DGLA, and DHA wt %**			
Variables	*t*-value	*p*	adjusted *R*^2^
LiA	−1.61	ns	0.160
DGLA	−4.72	**<0.0001**	
DHA%	3.96	**<0.0001**	

AIS: acute ischemic stroke; ns, not significant.

**Table 4 nutrients-12-02411-t004:** Threshold values for age of AIS onset using receiver operating curves from logistic regression analysis.

	*N*	Sens (%)	Spec (%)	PPV (%)	Odds ratio	*p*	AUC	AICc	BIC
Age <60 years as an OV									
DGLA (≥117.7 vs. <117.7) µmol/L	525	55.7	81.3	28.1	1.07 (1.05–1.09)	<0.0001	0.727	339	348
DHA wt % (≤3.7 vs. >3.7)%	525	68.9	80.2	31.3	0.36 (0.27–0.48)	<0.0001	0.792	315	323
Age <80 years as an OV									
DGLA (≥76.3 vs. <76.3) µmol/L	525	76	48.9	73.4	1.05 (1.03–1.07)	<0.0001	0.535	657	666
DHA wt % (≤3.8 vs. >3.8)%	525	32.8	82.1	77.2	0.75 (0.65–0.87)	<0.0001	0.600	668	677

AIS: acute ischemic stroke; AICc: corrected Akaike information criterion; AUC: area under the curve, BIC: Bayesian information criterion; DGLA: dihomo-gamma-linolenic acid; DHA: docosahexaenoic acid; OV: objective variable; PPV: positive predictive value; Sens: sensitivity, Spec: specificity.

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
