# Peer review of "Association of Serum Fatty Acids at Admission with the Age of Onset of Acute Ischemic Stroke"

_nutrients, 2020, doi:10.3390/nu12082411_

Round 1

Reviewer 1 Report

The author should rewrite the introduction part and highlight the importance and innovation of the present study compared to previous studies. There are many references have been omitted.

Major comments:

There are many subjective assertions in the introduction, which makes the rationale of the study insufficient. For example, the author introduced that “Serum FA levels were not examined before or at ischemic event onset.” However, this is a subjective assertion, since there are many studies had reported the relationship between fatty acid and ischemic stroke, such as “Serum fatty acids and incidence of ischemic stroke among postmenopausal women, Stroke. 2013 Oct;44(10):2710-7.”, “Serum fatty acids and ischemic stroke subtypes in middle- and late-onset acute stroke patients. Clinical Nutrition Experimental. Volume 22, December 2018, Pages 19-29”, “Serum fatty acids and the risk of stroke. Stroke. 1995 May;26(5):778-82.”, “Plasma fatty acid composition and incident ischemic stroke in middle-aged adults: the Atherosclerosis Risk in Communities (ARIC) Study.Cerebrovasc Dis. 2013;36(1):38-46. ”. Additionally, the author asserted that the association of n-3 PUFAs with the onset of AIS in young or middle-aged people remains unknown. However, this association had been investigated (Cerebrovasc Dis. 2013;36(5-6):388-93. Low serum n-3 polyunsaturated fatty acid/n-6 polyunsaturated fatty acid ratio predicts neurological deterioration in Japanese patients with acute ischemic stroke), and the median age of patients that included in the present study is 75 years old. The patients are not young or middle-aged people.   

Author Response

Major comments:

There are many subjective assertions in the introduction, which makes the rationale of the study insufficient.

For example, the author introduced that “Serum FA levels were not examined before or at ischemic event onset.” However, this is a subjective assertion, since there are many studies had reported the relationship between fatty acid and ischemic stroke, such as

 “6: Serum fatty acids and incidence of ischemic stroke among postmenopausal women, Stroke. 2013 Oct;44(10):2710-7.”,

 “8: Serum fatty acids and ischemic stroke subtypes in middle- and late-onset acute stroke patients. Clinical Nutrition Experimental. Volume 22, December 2018, Pages 19-29”,

“5: Serum fatty acids and the risk of stroke. Stroke. 1995 May;26(5):778-82.”,

“7: Plasma fatty acid composition and incident ischemic stroke in middle-aged adults: the Atherosclerosis Risk in Communities (ARIC) Study. Cerebrovasc Dis. 2013;36(1):38-46.”.

  1. Thank you for your comments and suggestions.

We have revised the sentences in the introduction on page 1, as follows: “The relationship between serum FA levels under fasting conditions and stroke has been reported [5-7]. However, few studies have reported on serum FA levels at the onset of ischemic events [8,9], which are typically examined under non-fasting conditions and associated with the dietary intake of TGs.”

Additionally, the author asserted that the association of n-3 PUFAs with the onset of AIS in young or middle-aged people remains unknown. However, this association had been investigated (Cerebrovasc Dis. 2013;36(5-6):388-93. Low serum n-3 polyunsaturated fatty acid/n-6 polyunsaturated fatty acid ratio predicts neurological deterioration in Japanese patients with acute ischemic stroke), and the median age of patients that included in the present study is 75 years old. The patients are not young or middle-aged people.   

  1. Thank you for your comments and suggestions.

We have revised sentences in the Introduction on page 1 as follows: “A low serum n-3 PUFA/n-6 PUFA ratio may be associated with neurological deterioration in patients with AIS [10], and low serum n-3 PUFA levels may also be associated with the onset of AIS.“

Reviewer 2 Report

The authors sought to investigate the associations between the serum concentrations and proportions of FAs in patients admitted for AIS and the age of its onset. I felt that a lot of effort was contributed to achieve this goal, and this manuscript is clear logic and well-written. However, it is mandatory to explain/correct the manuscript in some points.  

  1. Several important papers should be included in the introduction to provide sufficient background on the regulation of DHA and ARA in the central nervous system- for example, Journal of Neuroinflammation 15, 202 (2018); Frontiers in Neurology 10, 642 (2019); Molecular Neurobiology 57, 1085 (2020); NeuroMolecular Medicine 22, 278 (2020); Metabolites 2019, 9(3), 40.
  2. Any correction on the p-value?

Author Response

The authors sought to investigate the associations between the serum concentrations and proportions of FAs in patients admitted for AIS and the age of its onset. I felt that a lot of effort was contributed to achieve this goal, and this manuscript is clear logic and well-written. However, it is mandatory to explain/correct the manuscript in some points.  

  1. Several important papers should be included in the introduction to provide sufficient background on the regulation of DHA and ARA in the central nervous system- for example,

Journal of Neuroinflammation 15, 202 (2018);

[14].         Frontiers in Neurology 10, 642 (2019);

Molecular Neurobiology 57, 1085 (2020);

NeuroMolecular Medicine 22, 278 (2020);

[13].         Metabolites 2019, 9(3), 40.

  1. Thank you for your suggestions.

We have added the following information to the Introduction on page 2: “Dietary DHA has beneficial effects on neurodevelopment [13]. DHA and AA are rich in phospholipids and are found in the central nervous system [14].”

  1. Any correction on the p-value?

  1. Thank you for your comments.

We did not correct the p-value. R2 is adjusted.

Round 2

Reviewer 1 Report

These authors have addressed my concerns. The manuscript has been improved.

Author Response

Thank you for your comments.

Reviewer 2 Report

Since the concentration of DHA was significantly changed in this study, I feel authors should include several important papers in the introduction (line 34) to provide sufficient background on the neuroprotective effects of n-3 PUFA in the CNS, especially DHA. - for example, Journal of Neuroinflammation 15, 202 (2018);Molecular Neurobiology 57, 1085 (2020); Journal of Proteome Research 19 (6), 2236-2246 (2020). These papers are very important to support the conclusion of this manuscript. 

Author Response

Thank you for your comments.

We have added the following sentence in Introduction, on page 1, line 34:

"Previous studies have reported the neuroprotective effects of docosahexaenoic acid (DHA) [5-7]."

We have added the following papers to the references:

5.  Yang, B.; Li, R.; Michael Greenlief, C.; Fritsche, K.L.; Gu, Z.; Cui, J.; Lee, J.C.; Beversdorf, D.Q.; Sun, G.Y. Unveiling anti-oxidative and anti-inflammatory effects of docosahexaenoic acid and its lipid peroxidation product on lipopolysaccharide-stimulated BV-2 microglial cells. Journal of neuroinflammation 2018, 15, 202, DOI:10.1186/s12974-018-1232-3.

6.  Geng, X.; Yang, B.; Li, R.; Teng, T.; Ladu, M.J.; Sun, G.Y.; Greenlief, C.M.; Lee, J.C. Effects of Docosahexaenoic Acid and Its Peroxidation Product on Amyloid-β Peptide-Stimulated Microglia. Molecular Neurobiology 2020, 57, 1085-1098, DOI:10.1007/s12035-019-01805-4.

7.  Yang, B.; Li, R.; Liu, P.N.; Geng, X.; Mooney, B.P.; Chen, C.; Cheng, J.; Fritsche, K.L.; Beversdorf, D.Q.; Lee, J.C., et al. Quantitative Proteomics Reveals Docosahexaenoic Acid-Mediated Neuroprotective Effects in Lipopolysaccharide-Stimulated Microglial Cells. J. Proteome Res. 2020, 19, 2236–2246, DOI:10.1021/acs.jproteome.9b00792.